# Thyroid Hormone Augmentation for Bipolar Disorder: A Systematic Review

**DOI:** 10.3390/brainsci12111540

**Published:** 2022-11-14

**Authors:** Ashok Seshadri, Vishnu Sundaresh, Larry J. Prokop, Balwinder Singh

**Affiliations:** 1Department of Psychiatry & Psychology, Mayo Clinic, 200 First Street SW, Rochester, MN 55905, USA; 2Department of Psychiatry and Psychology, Mayo Clinic Health System, Austin, MN 55912, USA; 3Division of Endocrinology, University of Utah School of Medicine, Salt Lake City, UT 84108, USA; 4Mayo Medical Libraries, Mayo Clinic College of Medicine, Rochester, MN 55905, USA

**Keywords:** thyroid hormone, bipolar disorders, bipolar depression, rapid cycling, levothyroxine, LT4, liothyronine, T3

## Abstract

Thyroid hormone (TH) augmentation, although commonly used for major depression, is sparingly used for bipolar disorder (BD) after the failure of mood-stabilizing agents. While the exact mechanisms of thyroid hormone action in BD remains unclear, central thyroid hormone deficit has been postulated as a mechanism for rapid cycling. This systematic review—conducted in accordance with the PRISMA guidelines—of eight studies synthesizes the evidence for TH augmentation in BD. A systematic search of the Ovid MEDLINE, Embase, PsycINFO, and Cochrane databases was conducted for randomized controlled trials (RCT), open-label trials, and observational studies of levothyroxine (LT4) and triiodothyronine (T3) for BD. Open-label studies of high dose LT4 augmentation for bipolar depression and rapid cycling showed improvement in depression outcomes and reduction in recurrence, respectively. However, an RCT of high-dose LT4 did not show benefit in contrast to placebo. An RCT comparing LT4, T3, and placebo showed benefit only in rapid-cycling bipolar women. A meta-analysis could not be completed due to significant differences in study designs, interventions, and outcomes. Our systematic review shows mixed evidence and a lack of high-quality studies. The initial promise of supratherapeutic LT4 augmentation from open-label trials has not been consistently replicated in RCTs. Limited data are available for T3. The studies did not report significant thyrotoxicosis, and TH augmentation were well tolerated. Therefore, TH augmentation, especially with supratherapeutic doses, should be reserved for highly treatment-resistant bipolar depression and rapid-cycling BD.

## 1. Introduction

Bipolar disorder (BD) is a chronic condition characterized by recurrent episodes of mania/hypomania/depression or mixed states. Bipolar depression and the rapid cycling forms of illnesses are often challenging to treat due to a high degree of resistance to many standard mood stabilizers [1,2]. There are only five FDA-approved treatment options for bipolar depression—olanzapine + fluoxetine combination, quetiapine, lurasidone, cariprazine, and lumateperone, but there are none for treatment-resistant cases (Figure 1) [3]. The treatment of rapid-cycling bipolar disorder is largely guided by systematic reviews that recommend the withdrawal of antidepressants; evaluating possible precipitants (alcohol, stressors, thyroid dysfunction); optimizing mood stabilizer treatments including combining mood stabilizers; and considering adjunctive options such as atypical antipsychotic medications, anticonvulsants, and high-dose levothyroxine [4]. Thyroid abnormalities have long been suspected of playing a role in mood dysregulation, with common clinical observations of depression accompanying hypothyroidism. Some of the early published accounts of thyroid hormone (TH) augmentation for mood disorders date back to the 1950s with the discovery of the antidepressant activity of thyroid hormones, leading to the use of liothyronine (T3) augmentation for refractory depression [5,6,7]. An association among hypothyroidism, reduced central serotonergic activity, and depression was observed, along with the discovery that TH plays an important role in both serotonin and catecholamine functions in the brain [8]. Subsequent studies have reported an association between hypothyroidism and decreased brain metabolic activity that resolved after achieving euthyroid status with thyroid hormone therapy [9].

The utilization of T3 for treatment-resistant major depression has been an established clinical practice further cemented by the STAR*D study, that not only showed effectiveness comparable to lithium augmentation, but was also better tolerated [10]. A recent network meta-analysis showed both levothyroxine (LT4) and T3 as effective augmentation agents for major depression [11]. The systematic study of TH augmentation for BD began with the publication of a case series by Stancer and Persad [12], who reported that 5/8 women achieved remission from a rapid cycling course with supratherapeutic doses of LT4 (500 mcg per day) as an augmentation strategy for mood stabilizing treatment. Subsequently, several open-label studies and randomized controlled trials (RCTs) have examined TH augmentation with LT4 in moderate-to-high doses and T3 in the treatment of rapid cycling BD and bipolar depression [13,14,15,16,17,18]. The studies in BD have varied between using doses of LT4 (50–150 mcg per day) [19] to supratherapeutic doses (300–600 mcg per day) [13,14]. However, several open-label studies showed that only supratherapeutic doses offered benefits to reduce depression and rapid cycling in BD [12,13,14,18].

Thyroid hormone augmentation, although commonly used for major depression, is sparingly used for BD after the failure of mood stabilizing agents. This remains a lower tiered recommendation in practice guidelines such as the Canadian Network for Mood and Anxiety Treatments guidelines [20,21]. This is partly due to a smaller evidence base containing several open-label studies and few RCTs. A recent systematic review included studies of supratherapeutic LT4 augmentation for bipolar depression, but did not include T3 augmentation studies [22]. The clinical trials investigating the efficacy of high dose LT4 in bipolar depression did not report consistent results [14,15,16], and one of the RCTs investigating T3 augmentation for bipolar depression was negative [23]. The aim of our review is to systematically evaluate the literature to assess the role of LT4 and T3 augmentation in the treatment of BD, regardless of the mood state, and offer clinical guidance to medical practitioners.

## 2. Methods

### 2.1. Data Sources and Search Strategies

The databases, from the origin of the database to the latest entries, included Ovid MEDLINE (R) and Epub Ahead of Print; In-Process & Other Non-Indexed Citations; Daily; Ovid EMBASE; Ovid Cochrane Central Register of Controlled Trials; Ovid Cochrane Database of Systematic Reviews; Ovid PsycINFO; and Scopus. The search strategy was designed and conducted by an experienced librarian (LJP) with input from the principal investigator (BS). Controlled vocabulary supplemented with keywords was used to search studies for the therapeutic use of thyroid hormones, T3 and LT4, in BD. The actual strategy listing all search terms used and how they are combined is available in Appendix A.

A comprehensive search of several databases from each database’s inception to 16 September 2022, including all languages, was conducted with the following population, intervention, control, and outcomes terms: P = patients with BD; I = thyroid hormone therapy—liothyronine/triiodothyronine/T3 (Cytomel, Triostat); levothyroxine/thyroxine—T4 (Synthroid, Levoxyl, Tirosint, Unithroid, Thyquidity, Euthyrox); C = placebo/control; O = improvement in mood, depression, mania, or cycling. We followed the Preferred Reporting Items for Systematic Reviews and Meta-Analyses reporting guidelines [24].

The online systematic review tool, Endnote, was used for title and abstract screening and full-text review (Endnote, Version X9, 2019). Title and abstract screening were performed in duplicate by two independent reviewers (AS and BS). Conflicts were resolved by discussion.

### 2.2. Inclusion and Exclusion Criteria

Our inclusion criteria were RCTs, open-label clinical trials, and observational studies; conducted in adult patients with BD, in any phase of illness, such as rapid cycling or depression; compared against any control condition, including other treatment interventions or treatment as usual; in a current episode, diagnosed using standardized diagnostic criteria. We included conference presentation abstracts if they contained an adequate report of the study data and screened for unpublished studies as part of our comprehensive search strategy. We excluded case reports and case series from this report. The primary outcome measure was a change in depressive symptom severity for bipolar depression, measured using standardized rating scales between the beginning and end of the treatment intervention period; and a reduction in episodes or clinical morbidity such as hospitalization or number of episodes with rapid-cycling bipolar disorder.

### 2.3. Data Extraction

The full-text review was completed in duplicate from studies by two authors which met the inclusion criteria, with any disagreements resolved by discussion and reaching consensus (AS and BS). The risk of bias assessment was completed by two independent authors (BS and VS). Data were extracted by two independent reviewers, with any conflicts being resolved through discussion and consensus (AS and BS).

### 2.4. Quality Assessment/Risk of Bias

The Cochrane Collaboration’s risk of bias tool was used for assessing the risk of bias for RCTs [25]. The risk of bias was assessed for random sequence generation, allocation concealment, the blinding of participants and personnel, the blinding of outcome assessment, incomplete outcome data, selective reporting, and other biases. For open-label non-randomized studies/observation studies, we used the Methodological Index for Non-Randomized Studies [26,27]. Quantitative tests to assess publication bias were not performed due to the limited number of studies.

### 2.5. Meta-Analytic Techniques Consideration

We planned to extract treatment response data using reported rates of remission from selected studies and change in depression scores with the primary outcome measure where reported. The meta-analysis of proportions was planned using a random effects model to synthesize the weighted average proportions of remission from the selected studies, using the inverse variance method. Standardized mean differences with 95% confidence interval to compute effect sizes between treatment groups was planned where depression outcome measures were reported. We planned to use R studio, Version 4.0.5 software to conduct the meta-analysis (R version 4.0.5 [2021]). We planned to assess heterogeneity using the Cochran Q statistic, with *p* value < 0.10 on the Cochran Q test used as a cut-off to attribute heterogeneity to between-study factors, rather than by chance [28] We planned to use I² statistic to assess the contribution of between-study heterogeneity to the overall estimate of heterogeneity.

## 3. Results

Our search strategy yielded 625 articles after de-duplication and title and abstract screening, from which 19 studies were found to be eligible for full-text review (Figure 2). Of the full-text articles assessed for eligibility, eight studies (N = 311) were included for the systematic review. We identified one open-label trial (LT4) and one RCT (LT4/T3) of adjunctive TH augmentation in rapid cycling BD [13,17]. We found three open-label trials and one RCT of adjunctive high-dose LT4 for bipolar depression [14,15,16,18]. We identified one RCT of T3 for bipolar depression presented as a conference abstract; however, we were not able to find a full publication [23]. We found one retrospective study of adjunctive T3 in treatment-resistant BD II and BD NOS [29]. Among these, five studies were conducted in the United States (US) [13,15,17,23,29] and two in Germany [14,18], with one being a multi-site study (US and Germany) [16]. The characteristics of the included studies are presented in Table 1. Considering the various study designs including a retrospective study (n = 159), open-label studies without control groups (n = 46), and RCTs (n = 60 THs, n = 46 placebo) with different interventions and outcomes, as well as differences in study populations, we concluded that a meta-analysis was not feasible.

### 3.1. Clinical Scenario

#### 3.1.1. Rapid-Cycling Bipolar Disorder

Bauer and Whybrow [13] conducted an open-label trial of high-dose LT4 in 11 patients (10 females) with rapid-cycling BD refractory to a stable regimen of mood stabilizing medications (such as lithium and carbamazepine at therapeutic levels). The LT4 dose was increased by 50–100 mcg per day every 1–2 weeks as tolerated until clinical response was achieved at doses between 150–400 mcg per day. At completion of study, 10 of the 11 participants showed significant reductions in depressive symptoms, along with significant improvement in manic and hypomanic symptoms. Four out of ten patients entered a placebo crossover trial to assess the role of LT4 in the clinical response. There was no relationship between baseline thyroid status at study entry and clinical response. The authors noted a minimal occurrence of adverse effects, including mild resting tremor in one subject and a transient increase in anxiety and agitation in another. There was no evidence of thyrotoxic features such as tachycardia or weight loss.

Walshaw et al. [17] conducted an RCT comparing LT4 and T3 as an adjunctive treatment in lithium refractory rapid-cycling BD for reducing episodes of illness cycling. Thirty-two (60% females) treatment-resistant rapid-cycling BD patients were randomized into three treatment arms of LT4, T3, and placebo and followed for 16 weeks. LT4 dosages were titrated to achieve a free T4 index between 4.5 and 7.5 units or until TSH suppression was achieved (<0.1 units). For the T3 group, dosages were titrated until T3-resin uptake levels of 0.65–1.36 units were achieved. Using a Markov chain analysis, the study found that the LT4 group spent significantly less time depressed (*p* = 0.02) and in a mixed state (*p* = 0.03) compared to baseline, as well as significantly greater time in euthymia (*p* = 0.02). There were no significant changes from pre- to post-treatment in any of the mood states for the T3 and placebo groups.

#### 3.1.2. Bipolar Depression

Bauer et al. [18] conducted the first open-label trial of augmentation using supratherapeutic/supraphysiologic doses of LT4, compared to conventional antidepressant therapy for 12 (92% female) euthyroid and severely depressed patients with BD (baseline Hamilton Rating Scale for Depression [HDRS] score 26.6 ± 4.7) with a mean depression duration of 11.5 ± 13.8 months. LT4 was titrated to a mean dose of 482 ± 72 µg/day. At the end of 8 weeks, 5/12 (42%) patients showed ≥50% reduction in HDRS, while one had a partial response and six had no response. The authors noted the occurrence of mild tremors and sweating as the most common adverse effects. Tachycardia was observed in a few participants. There was no evidence of thyrotoxicosis or any major cardiac adverse effects.

Bauer et al. [14] conducted a prospective open-label study of high-dose LT4 (378.6 ± 90.2 µg/day) in 21 patients with treatment-resistant affective disorders (BD = 13). The patients were followed longitudinally over 51.4 ± 21.7 months to investigate clinical outcomes, as well as the safety of high-dose LT4. Patients in the BD group noticed a significant reduction in the recurrence of depression, measured using the morbidity index (*p* = 0.02). They noted the occurrence of tremor in one subject and a mild increase in resting heart rate in the overall study group. However, there were no changes in blood pressure, weight, or bone density throughout the observation period.

Bauer et al. [15] conducted an open-label trial comparing 10 female patients with bipolar depression (mean duration of current episode was 171 ± 125 days, baseline HDRS 23.2 ± 5.0) with 10 female healthy controls, on high dose LT4 (320 ± 42.1 µg/day), with the dose titration occurring every week: 100 µg in week 1, 200 µg in week 2, and 300 µg in weeks 3–7. If TSH suppression was not achieved, the LT4 dose was increased to 400 µg/day. All patients with BD exhibited a significant decline in depression scores (*p* < 0.001). Seven patients with bipolar depression were classified as responders, while three were classified as partial responders. This study did not report data on adverse effects.

Kelly et al. [29] published a retrospective study of T3 augmentation for TRBD with 159 patients (62% female, BD II-125, BD NOS-34), using mean T3 doses of 90.4 mcg (range of 13 mcg–188 mcg per day), with a follow-up of 20.3 ± 9.7 months. They reported evidence of clinical improvement measured by Clinical Global Impression—Improvement score (CGI-I), in both BD II (CGI-I-1.9± 1.2) and BD NOS (CGI-I-1.8± 1.2). The study found no significant concerns with worsening depression or switch to hypomania on T3. Gender differences in response were not noted. The most common adverse effect was hand tremors, which responded to dose reduction. Although bone density was not systematically assessed, three female patients were identified with osteoporosis, all of whom had multiple risk factors for osteoporosis. One subject developed a recurrence of atrial fibrillation at 125 mcg and responded to medication therapy, including a reduction in T3 dose.

Staam et al. [16] conducted a multi-center, double blind RCT with a fixed dose of supratherapeutic LT4-300 µg/day, adjunctive to mood stabilizer and/or antidepressant medication treatment for BD I/II depressed patients (mean baseline HDRS of 21.2) Thirty-one participants (52% female) were randomized to receive LT4 or placebo and followed for 6 weeks. Change in HDRS from baseline was the primary treatment outcome. While the overall difference between the two groups was not significant for improvement in depressive outcome measures, female patients showed a significant difference between the intervention and placebo groups. No serious adverse events were recorded during the study. Three patients discontinued LT4 due to adverse effects (mild thyrotoxicosis (1), switch into mania (1), and exanthema (1)).

Braga et al. [23] conducted an 8-week, double-blind placebo-controlled RCT of T3 augmentation for patients with bipolar depression. Six participants were randomized to receive T3 augmentation (dosage details are not available) and placebo, respectively. The baseline characteristics included HDRS 17.9 ± 4.8 and 42% female patients. This study did not find a significant difference in treatment response between groups for depressive outcomes.

### 3.2. Quality of Included Studies 

Table 2 highlights the quality assessment of the included studies. Four of the included studies used an open-label design [13,14,15,18]; thus, they were at a high risk of bias due to the lack of a control group and because of small sample sizes. We identified three RCTs, of which one study was only available as an abstract [23] and not available to extract complete data. The two remaining RCTs [16,17] did not provide information regarding random sequence generation, allocation concealment, the blinding of participants and personnel, or the blinding of outcome assessment; thus, they were at a moderate-to-high risk of bias. All studies used validated outcome measures. We identified one retrospective study with a moderate-to-high risk of bias [29].

## 4. Discussion

Rapid-cycling BD and TRBD represent challenging clinical conditions characterized by high morbidity and treatment resistance to standard therapies. While TH augmentation for recurrent major depression is a recognized treatment strategy, the use of TH augmentation for bipolar illness remains sparsely utilized. In this systematic review, we comprehensively summarize the available literature of TH augmentation strategies in BD.

We identified only two studies of TH augmentation for rapid-cycling BD [13,17]; one (n = 11) open-label study, using supratherapeutic doses of LT4 at doses between 150–400 mcg, and a placebo-controlled RCT, with a small sample (n = 32) of supratherapeutic T4 augmentation, showed evidence of significantly decreased time spent in depression or mixed states and increased time in euthymia. These findings suggest a positive role for augmentation with high dose adjunctive LT4 for reducing the illness burden of treatment-resistant, rapid-cycling bipolar illness. There was no significant difference between placebo and T3 augmentation for reducing the burden of rapid-cycling illness.

Several open-label studies [12,13,14,18] showed promise of supratherapeutic LT4 augmentation in reducing the morbidity of bipolar depression, including a reduction in recurrent episodes, as well as the improvement of depressive symptoms. The findings also suggested a greater role of TH augmentation in women than men based on treatment responses. An RCT of 300 mcg per day of LT4 compared to placebo appeared to confirm the superior efficacy in women in contrast to men for bipolar depression, although the overall sample (that included men) did not show evidence of significant superiority compared to placebo [16]. The authors proposed that supratherapeutic doses of LT4 may be necessary to achieve a treatment response [13,14,15,18]. This is supported by studies using PET imaging [15], which showed that clinical improvement in bipolar depression is associated with changes (from baseline) in metabolism in several brain regions (the right subgenual cingulate cortex, left thalamus, right amygdala, right hippocampus, right dorsal and ventral striatum, and cerebellar vermis), after treatment with supratherapeutic LT4. Support for T3 augmentation in BD is mixed, with positive clinical outcomes reported in a retrospective study in patients with BD II depression and BD NOS [29], while a small, unpublished, RCT did not find any difference between T3 and placebo [23]. Based on the available evidence, the role for T3 augmentation is less robust in contrast to LT4.

Thyrotropin-releasing hormone (TRH) has been used in research settings but has limited availability in the clinical world. Although we identified a placebo-controlled RCT of a single nocturnal dose of 500 mcg TRH (n = 10) compared to normal saline (n = 10), to evaluate the antidepressant response in 20 bipolar depression patients [30], we decided not to include it in the review, as it is not available for clinical practice. This study showed rapid and significant improvement in bipolar depression symptoms with TRH within 24 h compared to placebo (52% vs. 12%). One of the notable features of this sample was that the TRH group had higher baseline TSH (4.9 ± 1.2 mIU/mL) in contrast to placebo (2.1 ± 1.2 mIU/mL), suggesting the possibility that subclinical hypothyroidism in the TRH group may have contributed to a significantly improved clinical outcome [30].

### 4.1. Adverse Effects

Despite the high doses of LT4, patients did not show any severe adverse effects during the short duration of the study period. Cardiac parameters were observed to be stable and there were no reports of weight loss. Mild hand tremors were the most commonly reported side-effect. Even though some patients discontinued treatment due to adverse effects early on in treatment, high-dose LT4 was well tolerated by a majority of the patients.

### 4.2. Strengths and limitations

Our systematic review has several strengths, including the robust literature search with an experienced librarian, study selection and data extraction, and quality assessment in duplicate. Despite a comprehensive search of the major databases, we only found a small number of RCTs. The majority of the studies were conducted by one group of researchers without replication in larger cohorts from different continents. We did not identify good quality studies of LT4 at replacement doses (50–150 mcg/day) typically used to treat hypothyroidism in BD. The overall quality of the available evidence is low, considering that most of the included studies were open label, lacking a control group. The RCTs were of small samples and had different interventions, making it challenging to draw conclusions. While some of the studies have adequate follow-up to gauge treatment side-effects, it is not possible to know the long-term impact of high-dose TH augmentation, considering that most patients will need lifelong treatment for their mood disorders. We decided not to conduct a meta-analysis due to the small number of included studies, and instead conducted a comprehensive systematic review.

### 4.3. Clinical Recommendations

Patients with bipolar depression or rapid-cycling illness need to be assessed at baseline and periodically monitored for thyroid status while on TH augmentation. If baseline TSH is high and accompanied by low free T4 indicative of overt hypothyroidism, patients should be evaluated by the primary care provider and/or endocrinologist to replace with a weight-based dose of LT4, in order to achieve euthyroid status as soon as possible. If TSH is high with a normal fT4 indicative of subclinical hypothyroidism, LT4 replacement may be especially important, in the context of sub-optimally controlled mood disorder, although the evidence base in bipolar depression is limited. The decision to use supratherapeutic doses of LT4 should be made in close collaboration with the patient, primary care physician, and/or endocrinologist. This should be reserved for patients with severe TRBD as an adjunctive treatment with diligent monitoring. Monitoring for adverse effects needs to include baseline and periodic assessment for symptoms and signs of thyrotoxicosis (palpitations/tachycardia, hand tremors, anxiety, insomnia, unintentional weight loss, oligo/amenorrhea, fragility fractures/low bone density) [21]. Long-term supratherapeutic TH augmentation, especially in postmenopausal women with reduced physical activity, a personal history of cigarette smoking, an excessive history of serotonergic antidepressant exposure, a family history of osteoporosis, and advanced age is associated with high risk for fragility fractures and osteoporosis [27]. T4 is often started at 50 mcg per day and optimized to 100–150 mcg per day [21]. Higher doses of LT4 should be carefully titrated based on clinical response and the achievement of adequate free T4 levels to produce TSH suppression.

### 4.4. Implications for Research

Most of the available evidence for LT4 augmentation for bipolar depression comes from a single research group who have conducted multiple open-label studies and a small RCT, over a span of 30 years. Considering the severity of disease burden associated with rapid-cycling BD and TRBD, and the relative frequency of suboptimal outcomes with standard mood-stabilizing therapies, the potential role of TH augmentation is promising. There is a critical need to conduct well-designed RCTs in patients with TRBD and rapid-cycling BD. The available evidence suggests that patients with more severe depression may benefit from TH augmentation compared to those with moderate symptoms, and females may have a greater response compared to males. These findings need to be further confirmed in treatment trials. The case for using supratherapeutic doses of LT4 stems from open-label case series and trials. There are no RCTs comparing normal-dose LT4 (100–150 mcg/day) with high-dose LT4 (300–500 mcg/day) in BD. In a small study, Bauer et al., using PET imaging, reported an improvement in clinical depression outcomes correlating with changes in cerebral metabolic activity, which was achieved using supratherapeutic LT4 augmentation [15]. Comparative PET imaging studies using replacement-dose LT4 versus supratherapeutic LT4 in this patient population may shed more light on the possible differences of brain metabolism induced by the two strategies.

## 5. Conclusions

Thyroid hormones play a pivotal role in mood regulation, and TH augmentation needs to be carefully assessed in the treatment of mood disorders. Bipolar depression and rapid-cycling BD represent conditions with a high degree of treatment resistance and clinical morbidity. The assessment of thyroid status and the adequate correction of abnormal thyroid status is paramount in achieving mood stabilization. The support for T3 augmentation in bipolar depression and rapid-cycling BD is mixed and needs further research. The current evidence for using supratherapeutic doses of LT4 based on open-label studies and one RCT shows promise in bipolar depressed women with severe illness and significant treatment resistance to multiple medications. Limited evidence suggests promise for the use of supratherapeutic doses of LT4 in rapid-cycling illness, in order to reduce the time spent in depressed or mixed states and increase time spent in euthymia. With limited therapeutic interventions for TRBD and rapid cycling, there is a need for urgent research. The likelihood of identifying an effective intervention is higher if the interventions are concurrently evaluated. Thus, an ideal way to examine the real effect of TH augmentation would be to conduct a large, multi-center, multiple-arms (high-dose LT4, normal-dose LT4, T3, and placebo) double-blind RCT, in patients with moderate-to-severe TRBD.

## Figures and Tables

**Figure 1 brainsci-12-01540-f001:**
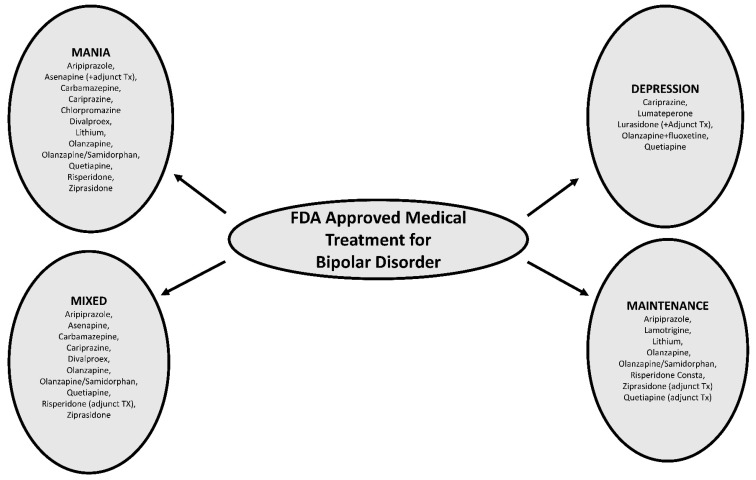
The FDA-approved pharmacotherapeutic medications for bipolar disorder. (Dexmedetomidine was recently FDA approved for agitation in bipolar disorder).

**Figure 2 brainsci-12-01540-f002:**
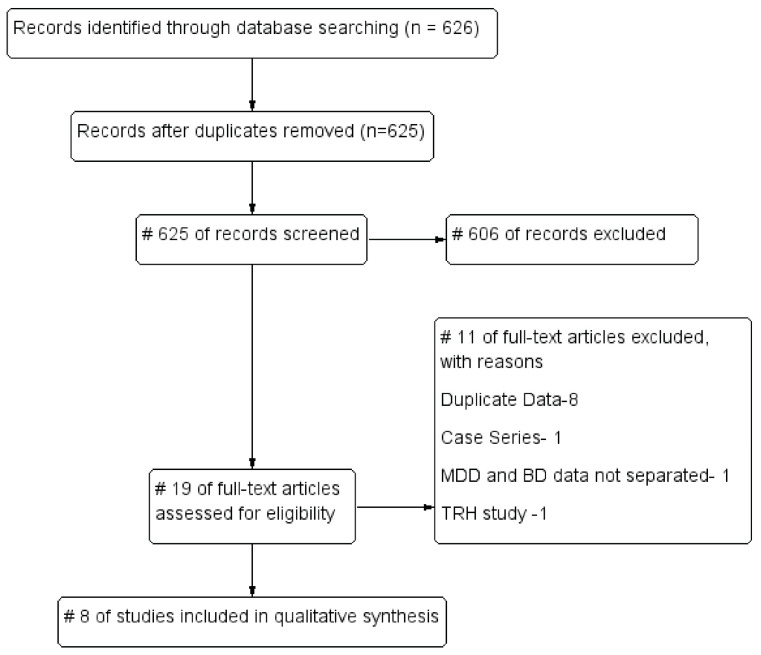
The study flow chart showing the study identification and selection.

**Table 1 brainsci-12-01540-t001:** Characteristics of the included studies.

Author/Year	Diagnosis	Study Design	Intervention	Participant Numbers—BP Only (Female %)	Duration	Outcome	Adverse Effects	Results/Comments
Rapid Cycling Bipolar disorder								
Bauer and Whybrow 1990, USA	Rapid-Cycling BD	Open Label (followed by placebo crossover trial n = 4)	LT4-150-400 mcg/day	11 (F = 91%)	Minimum of 60 days	HDRS, YMRS	Tremors = several (but not requiring treatment) Exacerbation of pre-existing tremors: 2 Anxiety:1 Other hyperthyroid symptoms: 0	10 of 11: Significant reduction in depression 5 of 7: Significant improvement in mania/hypomania
Walshaw, 2018, USA	Rapid Cycling	RCT	LT4, T3, and placebo LT4 dose titrated for a fT4 index 4.5–7.5 or TSH < 0.1 units T3 dose titrated for a T3 resin uptake 0.65–1.36 units	32 (F = 69%) LT4:13 (F = 62%) T3: 10 (F = 60%) Placebo: 9 (F = 89%)	≥16 weeks	Time spent in episode	LT4 and T3 group Mild tremors: 10 Diarrhea: 8 Hot flashes/sweating: 5 Mild palpitations, tachycardia: 4 Dizziness: 3	LT4: Less time depressed, increased time euthymic T3: No significant change Placebo: No significant difference
Bipolar Depression								
Bauer 1998, Germany	Bipolar depression	Open Label	LT4: 482 ± 72 mcg/day	12 (F = 92%)	≥8 weeks	HDRS ≤ 9 or ≥50% reduction	Sweating: 6 Tremors: 2 Tachycardia: 1	5 of 12 patients responded 6 of 12 had no response 1 of 12 had partial response
Bauer 2002, Germany	Resistant mood disorders incl BD-I and BD-II	Open Label-Prospective	LT4: 378.6 ± 90.2 µg/day	13 (F = 62%)	Longitudinal: 51 ± 21 months	Recurrence measured with morbidity index, Thyroid profile	Worsening of pre-existing tremor: 1 No significant change in heart rate, BP, weight, and bone density	Significant reduction in morbidity index in BD subjects (*p* = 0.02)
Bauer 2005, USA	Bipolar depression	Open Label	LT4-320 ± 42.1 µg/day	10 (F = 100%)	7 weeks	HDRS, BDI, CGI, PET imaging	Significant decrease in systolic BP No other significant adverse effects	7 of 10 responded 3 of 10 partial responders
Kelly 2009, USA	Bipolar depression	Retrospective Study	T3 Mean—90.4 mcg/day (range 13 mcg–188 mcg)	159 (F = 62.5%) (Bipolar II-125 Bipolar NOS-34)	20.3 ± 9.7 months	CGI-I	10% discontinued T3 due to adverse effects Tremor: Most common Osteoporosis: 3 (not systematically assessed)	BD II-84% Improved, 32%—Remission BD NOS-85% Improved, 38%—Remission
Braga 2013-Conference Abstract	Bipolar depression	RCT	T3 Dose not reported	12 (T3-6, Placebo-6)	8 weeks	HDRS	Not reported	No significant difference between T3 and Placebo
Staam 2014, Germany &USA	Bipolar depression	RCT	LT4 300 mcg/day	62 (F = 52%) LT4-31 (F = 55%) Placebo-31 (F = 48%)	6 weeks	HDRS Response = ≥50% reduction in HDRS. Remission = HDRS score ≤ 7	No serious adverse events LT4 discontinued in 3 patients (mild thyrotoxicosis, exanthema, and switch into mania) Inner restlessness: Number not reported No other hyperthyroid side effects	Response rate— LT4 vs. Placebo =36% vs. 26%, *p* = 0.41. Remission rates– LT4 vs. Placebo = 23% vs. 16%, *p* = 0.52. Secondary analysis in women showed significant reduction in mean HDRS scores (*p* = 0.02) but not in men.

BD = bipolar disorder; BP = blood pressure; CGI = Clinical Global Impressions Scale; CGI-I = Clinical Global Impression—Improvement; HDRS = Hamilton Depression Rating Scale; PET = positron emission tomography; YMRS = Young Mania Rating Scale.

**Table 2 brainsci-12-01540-t002:** (A): Risk of bias for RCTs included in the systematic review. (B): Quality assessment of the open-label studies included—Methodological Index for Non-Randomized Studies (MINORS).

(A)									
Criteria		Braga, 2013 [23]		Staam, 2014 [16]			Walshaw, 2018 [17]		
Random sequence generation (selection bias)		Unclear risk		Unclear risk			Unclear risk		
Allocation concealment (selection bias)		Unclear risk		Unclear risk			Unclear risk		
Blinding of participants and personnel		Unclear risk		Unclear risk			Unclear risk		
Blinding of outcome assessment (detection bias)		Unclear risk		Unclear risk			Unclear risk		
Incomplete outcome data addressed (attrition bias)		Unclear risk		Low risk			Low risk		
Selective reporting (reporting bias)		Unclear risk		Low risk			Low risk		
Other bias		Unclear risk		Unclear risk			Unclear risk		
(B)									
Methodological Items for Non-Randomized Studies			Score †						
			Bauer, 1990 [13]		Bauer, 1998 [18]	Bauer, 2002 [14]		Bauer, 2005 [15]	Kelly, 2009 [29]
1	A clearly stated aim: the question addressed should be precise and relevant in light of the available literature.		2		2	2		2	1
2	Inclusion of consecutive patients: all patients potentially fit for inclusion (satisfying the criteria for inclusion) have been included in the study during the study period (no exclusion or details about the reasons for exclusion).		2		2	2		2	2
3	Prospective collection of data: Data were collected according to a protocol established before the beginning of the study.		2		2	2		2	1
4	Endpoints appropriate to the aim of the study: unambiguous explanation of the criteria used to evaluate the main outcome, which should be in accordance with the question addressed by the study. Moreover, the endpoints should be assessed on an intention-to-treat basis.		2		2	2		2	1
5	Unbiased assessment of the study endpoint: blind evaluation of objective endpoints and double-blind evaluation of subjective endpoints. Otherwise, the reasons for not blinding should be stated.		0		0	0		0	0
6	Follow-up period appropriate to the aim of the study: the follow-up should be sufficiently long to allow the assessment of the main endpoint and possible adverse events.		2		2	2		2	0
7	Loss to follow-up less than 5%: all patients should be included in the follow up. Otherwise, the proportion lost to follow-up should not exceed the proportion experiencing the major endpoint.		2		2	2		2	2
8	Prospective calculation of the study size: information of the size of detectable difference of interest with a calculation of 95% confidence interval, according to the expected incidence of the outcome event, and information about the level for statistical significance and estimates of power when comparing the outcomes.		0		0	0		0	0

† The items are scored 0 (not reported), 1 (reported but inadequate) or 2 (reported and adequate), the global ideal score being 16 for non-comparative studies.

## Data Availability

Data are contained within the article or Appendix A.

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
