# Peer review of "Thyroid Hormone Augmentation for Bipolar Disorder: A Systematic Review"

_brainsci, 2022, doi:10.3390/brainsci12111540_

Round 1

Reviewer 1 Report

This is a very interesting review focused on thyroid hormone augmentation in patients suffering from Bipolar Disorder. The paper is well-written and of interest for the journal, and opens new avenues on the research of this important topic. However, I consider that the paper needs several minor changes before considering it for publication.

Abstract.

1- A short sentence explaining why this research is important. Is there any mechanism explaining the hypothesis of efficacy of Thyroid hormone in Bipolar disorder?

2-This systematic review was done according to the PRISMA criteria. This should be described in the abstract section. 

Introduction

1- In the introduction section the authors report that thyroid hormone can be used in treatment resistant cases or case difficult to treat such as rapid cycling. What other treatments are used in that cases? I consider that it should be mentioned before describing more in depth why is thyroid hormone used.

2-The present review aims to systematically sumarize the literature evaluating the role of LT4 and T3 supplementation in the treatment of Bipolar Disorder. I recommend to expand this section of Aims and Objectives.

Methods

1- Subsection 2.2, should be renamed as "Inclusion and Exclusion criteria". A previous subsection on the screening and selection processes should be built.

Results

1- In the results section the authors are describing two main clinical scenarios: Bipolar depression and Rapid cycling. I recommend to joint both into a unique section called: clinical scenarios, and 3.2. could be renamed as 3.3.: quality of studies.

Discussion

1-The vast majority of studies are open label, and their results are not seemingly replicated in RCTs. This is a limitation that the authors have explained in the 4.2. section. Which other clinical scenarios and other study designs are the authors proposing for future research?

2- The subsection 4.3. Clinical Practice should be renamed as Clinical Recommendations.

Author Response

Reviewer 1.

This is a very interesting review focused on thyroid hormone augmentation in patients suffering from Bipolar Disorder. The paper is well-written and of interest for the journal, and opens new avenues on the research of this important topic. However, I consider that the paper needs several minor changes before considering it for publication.

Abstract.

1- A short sentence explaining why this research is important. Is there any mechanism explaining the hypothesis of efficacy of Thyroid hormone in Bipolar disorder?

Thank you. We have added the following statement –

“While the exact mechanisms of thyroid hormone action in bipolar disorders remains unclear, central thyroid hormone deficit have been postulated as a mechanism for rapid cycling.”

2-This systematic review was done according to the PRISMA criteria. This should be described in the abstract section. 

Thank you. We have added this.

Introduction

1- In the introduction section the authors report that thyroid hormone can be used in treatment resistant cases or case difficult to treat such as rapid cycling. What other treatments are used in that cases? I consider that it should be mentioned before describing more in depth why is thyroid hormone used.

Thank you.

We have added the following – “There are only five FDA approved treatment options for bipolar depression – olanzapine+fluoxetine combination, quetiapine, lurasidone, cariprazine, and lumateperone, although none for treatment resistant cases. Treatment of rapid cycling bipolar disorder is largely guided by systematic reviews that recommend withdrawal of antidepressants, evaluating possible precipitants (alcohol, stressors, thyroid dysfunction), optimizing mood stabilizer treatments including combining mood stabilizers and considering adjunctive options such as atypical antipsychotic medications, anticonvulsants, and high-dose levothyroxine.”

2-The present review aims to systematically summarize the literature evaluating the role of LT4 and T3 supplementation in the treatment of Bipolar Disorder. I recommend to expand this section of Aims and Objectives.

Thank you. We have added the following -

The clinical trials investigating efficacy of high dose LT4 in bipolar depression did not report consistent results [13-15] and one of the RCTs investigating T3 augmentation for bipolar depression was in fact negative [22]. The aim of our review is to systematically evaluate the literature to evaluate the role of LT4 and T3 augmentation in the treatment of BD, regardless of the mood state, and offer clinical guidance to medical practitioners.

Methods

1- Subsection 2.2, should be renamed as "Inclusion and Exclusion criteria". A previous subsection on the screening and selection processes should be built.

Thank you. Done.

Results

1- In the results section the authors are describing two main clinical scenarios: Bipolar depression and Rapid cycling. I recommend to joint both into a unique section called: clinical scenarios, and 3.2. could be renamed as 3.3.: quality of studies.

Thank you. Done.

Discussion

1-The vast majority of studies are open label, and their results are not seemingly replicated in RCTs. This is a limitation that the authors have explained in the 4.2. section. Which other clinical scenarios and other study designs are the authors proposing for future research?

Thank you. The likelihood of identifying an effective intervention is higher as the interventions are evaluated concurrently. Thus, an ideal way to examine the real effect would be to conduct a large multicenter, multiple arms (high dose LT4, normal dose LT4, T3, and placebo) double blind randomized controlled trial in patients with TRBD.

2- The subsection 4.3. Clinical Practice should be renamed as Clinical Recommendations.

Thank you. Done.

Reviewer 2 Report

The current systematic review presents a solid methodology and discussion. Although very few studies were included, this was the consequence of a thorough database search and study selection, and the limitations are described more than once by the authors, in a critical analysis manner. Nevertheless, I advise for a few modifications before acceptance for publication:

-        - Improve the quality of figure 1, “Flow diagram”, and also its caption (better, more thorough description of what is depicted);

-        - Table 2 B has numbers colored in blue, remove the coloring;

-        - Using so many abbreviations becomes confusing for the reader, you should write the full words whenever you can;

-        - In section title “3.2. Bipolar Depression:” remove the “:” at the end;

-        - In the introduction section do a figure regarding the pathophysiology and current treatment of bipolar disorder, for better understanding;

-        - In the conclusion section, suggest what could/should be done in order to uniformize the protocols of clinical trials, for them to be more easily directly comparable.

Author Response

The current systematic review presents a solid methodology and discussion. Although very few studies were included, this was the consequence of a thorough database search and study selection, and the limitations are described more than once by the authors, in a critical analysis manner. Nevertheless, I advise for a few modifications before acceptance for publication:

-        - Improve the quality of figure 1, “Flow diagram”, and also its caption (better, more thorough description of what is depicted);

Thank you. We have updated the flow diagram. 

-        - Table 2 B has numbers colored in blue, remove the coloring;

Thank you. Done.

-        - Using so many abbreviations becomes confusing for the reader, you should write the full words whenever you can;

Thank you. We have removed THA and RCBD.

-        - In section title “3.2. Bipolar Depression:” remove the “:” at the end;

Thank you. Done.

-        - In the introduction section do a figure regarding the pathophysiology and current treatment of bipolar disorder, for better understanding;

Thank you for the suggestion. We have added a figure (Figure-1) highlighting current FDA approved treatments for bipolar disorder. It is difficult to produce one figure which could explain the pathophysiology of bipolar disorder and we are not sure if that would add anything extra to the review. Thus, we kindly decided against creating a figure for pathophysiology.

-        - In the conclusion section, suggest what could/should be done in order to uniformize the protocols of clinical trials, for them to be more easily directly comparable.

Thank you. The likelihood of identifying an effective intervention is higher if the interventions are evaluated concurrently. Thus, an ideal way to examine the real effect of TH augmentation would be to conduct a large multicenter, multiple arms (high dose LT4, normal dose LT4, T3, and placebo) double-blind RCT in patients with moderate-severe TRBD.
